# Genome Analysis of Nicotinamide Adenine Dinucleotide-Independent *Mycoplasma synoviae* Isolates from Korea

**DOI:** 10.3390/pathogens10101231

**Published:** 2021-09-23

**Authors:** Yongjun Song, Tae-Min La, Taesoo Kim, Gyuhee Ahn, Won Hur, Hong-Jae Lee, Hyunjin Shin, Seung-un Song, Eunjin Park, Joong-Bok Lee, Seung-Yong Park, In-Soo Choi, Sang-Won Lee

**Affiliations:** College of Veterinary Medicine, Konkuk University, Seoul 05029, Korea; thedrakesng@gmail.com (Y.S.); fkxoals@konkuk.ac.kr (T.-M.L.); taesoo958111@konkuk.ac.kr (T.K.); pp0179@konkuk.ac.kr (G.A.); wonster8@konkuk.ac.kr (W.H.); lhj90@konkuk.ac.kr (H.-J.L.); shj1051@konkuk.ac.kr (H.S.); blankdo0702@konkuk.ac.kr (S.-u.S.); peunjin96@konkuk.ac.kr (E.P.); virus@konkuk.ac.kr (J.-B.L.); paseyo@konkuk.ac.kr (S.-Y.P.); ischoi@konkuk.ac.kr (I.-S.C.)

**Keywords:** chicken, *Mycoplasma synoviae*, nicotinamide adenine dinucleotide, NAD, comparative analysis

## Abstract

*Mycoplasma synoviae* (MS) is an avian pathogen that causes respiratory disease, infectious synovitis, and eggshell apex abnormalities in chickens. Nicotinamide adenine dinucleotide (NAD)-independent MS was first reported in 1975. Despite the atypical traits of NAD-independent MS, its independence from NAD has not been studied. In this study, we isolated five NAD-independent strains from Korea and assembled their genomes using sequencing reads obtained from Illumina and Oxford Nanopore Technology platforms. The assembled genomes were compared with the genomes of MS-H vaccine strain and type strain WVU1853. We found that the coding sequences of nicotinate phosphoribosyltransferase and glycerol-3-phosphate acyltransferase, and a unique coding sequence were present only in the genomes of NAD-independent isolates.

## 1. Introduction

*Mycoplasma synoviae* (MS) is an avian pathogen that causes respiratory disease, infectious synovitis, and eggshell apex abnormalities in chickens [1]. Ever since it was first reported in 1954 by Olson et al. [2], many studies have been conducted to isolate and incubate MS. After many attempts to design and improve the artificial medium for incubating MS, Chalquest [3], Frey et al. [4], and Olson and Meadow [5] established the use of nicotinamide adenine dinucleotide (NAD) for the in vitro growth of MS.

MS strains that do not require NAD for growth have also been reported several times. In 1975, DaMassa and Adler reported MS strains that adapted to media supplemented with nicotinamide instead of NAD [6]. In 1984, Yagihashi and Kato also discovered that MS can grow in an NAD-free broth [7]. However, although NAD-independent MS1 vaccine strain is now commercially available [8], the metabolic pathway that enables MS to grow without NAD remains unknown.

In this study, our lab isolated MS strains that do not require NAD. To understand the NAD independence of the MS isolates, we performed comparative genome analyses on the isolates using their newly determined complete genomes. The Illumina and Oxford Nanopore Technology (ONT) sequencing were performed to determine the complete genomes. Comparative analyses were performed to discover the unique coding sequences (CDSs) present on NAD-independent MS strains.

## 2. Results

### 2.1. Confirmation of the NAD-Independence of MS Isolates from Korea

Five MS strains capable of growing in Modified Frey’s Broth (MFB) without NAD were isolated. To confirm their NAD-independence, the MS isolates, type strain WVU1853 and vaccine strain MS-H were incubated in MFB without NAD supplement and with NAD (Sigma-aldrich, Saint Louis, MO, USA) supplement, respectively. Color-changing unit (CCU) was counted to measure the growth efficacy of each broth [9].

The growth of MS isolates obtained from Korea in NAD-negative broth was measured (Table 1). The MS isolates could grow in the MFB without NAD supplements. Their CCU ranged from 9.40 × 10^5^ to 30.1 × 10^6^ while neither WVU1853 nor MS-H strain grew in MFB without NAD. The ratio between the CCU in MFB without NAD supplement and CCU in MFB with NAD supplement varied from 0.86 to 6.76.

### 2.2. Complete Genome Sequencing of the MS Isolates from Korea

Five newly isolated MS strains and two complete genomes (MS-H and WVU1853) from GenBank were used in this study. Both the short-read sequences generated by Illumina sequencing and the long-read sequences generated by ONT sequencing were used to obtain the complete genome sequences of the MS isolates.

The Illumina and ONT sequencing reads were obtained as the result of each sequencing methods. An average of 5.4 million reads containing 1.6 billion bases were retrieved following Illumina sequencing. The Illumina sequencing of strain A4 was performed prior to the other strains. The percentage of bases with a quality score of 30 or higher (>Q30%) among all strains were above 89%. The number of ONT reads varied from 59,773 to 870,905 with an average of 494,851 reads. The average yield of the bases was 2,303,609,263 bp. The percentage of bases with a quality score of 10 or higher (>Q10%) of the ONT reads varied from 76.6% to 88.4%. The N_50_ of the reads ranged from 5000 to 16,143. Detailed information is available in the Appendix A.

The ONT sequencing read of each strain was de novo assembled using Flye long read assembler [10] and then polished with Illumina reads using unicycler-polish, which is part of the Unicycler pipeline [11]. Assembly of each strain yielded a single, circular contig (Table 2 and Appendix A).

### 2.3. General Characteristics and Annotation of the Assembled MS Genomes

The assembled genomes (18DW, 51SH, BS4S2, G3, and A4) and the genomes of MS-H [12] and WVU1853 [13] from GenBank were annotated using the Prokka, a genome annotation tool [14]. The average length of the assembled genomes was 795,375 bp. Each genome contained 34 tRNA, 1 tmRNA, 7 rRNA, 2 ncRNA and CDSs ranging from 672 to 789 sequences including pseudogenes (Table 3). The number of CDSs ranged from 640 to 731 when the pseudogene region was excluded. Detailed information is available in the Appendix A.

### 2.4. Comparative Genome Analysis of NAD-Independent and NAD-Dependent MS Strains

#### 2.4.1. Comparison of Genomes of MS Strains

BLAST Ring Image generator (BRIG) [15] was used to visualize the MS genomes. 18DW, 51SH, BS4S2, G3, A4, MS-H and WVU1853 strains were used with the genome of 18DW set as a reference. The differences in genomes of NAD-independent MS strains and NAD-dependent MS strains were identified and are shown in Figure 1.

#### 2.4.2. Analysis Using Roary and MegaBLAST

Roary, a pan-genome pipeline [16], was used to acquire additional clues explaining the NAD-independence of MS, with default parameters. Roary results showed the ten CDSs that were present only in the genomes of NAD-independent MS strains. MegaBLAST of each CDS were performed to recheck their presence on other MS strains, especially MS-H and WVU1853. The CDSs of 18DW were used as the representative while performing a BLAST search. MegaBLAST search was performed using the nr/nt database. Three coding sequences out of the ten CDSs from the Roary output were found to be absent on the genomes of MS-H and WVU1853.

A 291-bp long hypothetical protein CDS, which could not be annotated by the Prokaryotic Genome Annotation Pipeline (PGAP), was present on the genomes of the Chinese MS isolate, HN01 [17] and FJ-01 with query coverage of 100% and pairwise identity of 99.7% on both genomes.

A 1011 bp-long nicotinate phosphoribosyltransferase (NAPRT) CDS was found through the BLAST search. NAPRT is known to convert nicotinic acid to nicotinic acid mononucleotide (NaMN). It was detected on the genomes of HN01 and FJ-01 MS strain genomes. Both query coverage and pairwise identity were 100% on both strains. Additionally, NAPRT was present on the genome of Mycoplasma sp. NEAQ87857 with a query coverage of 38.48% and pairwise identity of 72.0%.

PlsY_2, which is 306 bp long, was matched with glycerol-3-phosphate acyltransferase (GPAT) CDS in the HN01 and FJ-01 strains. GPAT is an enzyme involved in lipid metabolism. Its query coverage was 100% and pairwise identity was 99.7% in both MS strains (Table 4). Detailed information is available in the Appendix A.

#### 2.4.3. Reannotation by BlastKOALA and Reconstruction of Metabolic Pathways

The annotation output files of Prokka in the faa format were reannotated using BlastKOALA [18] to fit the internal format of the Kyoto encyclopedia of genes and genomes (KEGG) pathway. For strain 18DW, 716 queries were taken as entries and 399 of the entries were re-annotated using BlastKOALA. The metabolic pathways were reconstructed using the annotated amino acid sequences by the KEGG mapper [19], reconfirming that, unlike WVU1853 and MS-H strains, NAPRT (Enzyme Commission number; EC 6.3.4.21) is present in the genomes of NAD-independent MS strains from Korea (Figure 2). NaMN, the metabolite of NAPRT enzyme is later converted to nicotinic acid adenine di nucleotide (NaAD) by nicotinate-nucleotide adenylyltransferase (NMNAT, Enzyme Commission number EC 2.7.7.18). The discovery of NAPRT in the genomes of NAD-independent MS strains indicates the NAD-independent strains are capable of synthesizing deamino-NAD from nicotinate.

## 3. Discussion

Genomic data of the NAD-independent strains of MS have been obtained through this study. Ever since MS was first reported in 1954 [2], only seven genomes of different strains of MS were available on the National Center for Biotechnology Information (NCBI) database. Now, there are 12 genomes of MS that are available from the NCBI database.

MS strains that do not require NAD have been reported several times in the past. Moreover, MS1 vaccine was recently developed. NAD-independence of such strains is of interest to both research labs and industries; however, it has not been studied yet. Therefore, no studies on the comparative genome analysis of the NAD-dependent and NAD-independent MS strains have been published till date.

The comparative genome analysis of the NAD-independent genomes revealed the unique CDSs present only in NAD-independent MS strains, *viz.*, hypothetical CDS, nicotinate phosphoribosyltransferase CDS, and glycerol-3-phosphatase acyltransferase CDS. These CDSs were also found in the genomes of the MS strains HN01 [17] and FJ-01 from China. It is not known whether these two strains are NAD-independent. However, if they are NAD-independent, then this independence may be a common trait shared among the MS strains of Eastern Asia, along with an NAD-independent strain discovered by Yagihashi et al. in Japan [7].

NAPRT on the “nicotinate and nicotinamide metabolism” section of KEGG was located using KEGG mapper. The complete biosynthetic pathway to synthesize the deamino-NAD was revealed. However, the pathway to synthesize NAD is not revealed through this study. Metabolic pathways suggested by the KEGG does not impeccably represent the metabolic pathway of the bacteria. For instance, KEGG cannot describe the metabolic pathway of NAD in *Francisella tularensis*. A new role of NH_3_-dependent NAD synthetase (NadE) enzyme of the *F.tularensis* was discovered by Sorci et al. in 2009 [20]. NadE was previously known to convert NaAD to NAD. Sorci et al. discovered that NadE of *F.tularensis* can convert NaMN to nicotinamide mononucleotide. However, this discovery is not indicated in the nicotinate and nicotinamide metabolism map of the KEGG pathway. Since the metabolism data available in KEGG are not complete, we expect the de novo pathway for synthesizing NAD to be present on the genomes of the NAD-independent strains.

Glycerol-3-phosphate acyltransferase and a unique CDS were studied yet they were thought to be irrelevant to NAD metabolism. The genomic regions that vary for each strain were not studied since they were not shared within the NAD-independent or NAD-dependent group.

The MS1 vaccine, which is well known to be NAD-independent, is presumed to have a different mechanism for its NAD-independent nature. The MS1 vaccine was developed by in vitro passaging of type strain ATCC 25204 (WVU1853) under laboratory conditions [8]. Because this type of strain is a parent strain of the MS1 vaccine, NAPRT or GPAT will not be present in the genome of the MS1 vaccine. Neither gene induction nor insertion is likely to occur under the laboratory conditions, which suggest that NAPRT or GPAT will not be present on the MS1 genome.

The NAD-independency of MS is an atypical trait that may reduce the cost of incubation in the industry and can bring interest to the researchers. However, since this study is only based on the in silico data, in vitro studies such as proteomics and metabolomics are needed.

## 4. Materials and Methods

### 4.1. MS Strains Used in This Study

Each MS isolated from farms in Korea from 2017 to 2020, MS-H vaccine (Vaxsafe MS, Bioproperties, Australia) and WVU1853 (ATCC 25204) were also used in this study. Fasta files from accession number CP021129 and CP011096 were used to retrieve the genomes of MS-H [12] and WVU1853 [13], respectively (Table 5).

### 4.2. CCU Count of MS Strains in MFB with and without NAD

MFB with and without 0.01% NAD supplement (Sigma-aldrich, Saint Louis, MO, USA) were used to incubate the MS isolates from Korea, strain WVU1853 and MS-H vaccine. MS stocks were incubated in the MFB with NAD until the logarithmic phase of growth was reached. The MS isolates were then centrifuged at 10,000× *g* for 10 min in 4 °C and suspended in 1 mL of phosphate buffered saline. Next, 25 µL of suspension were dispensed into each well of the first column of four different 96-well plates containing 225 µL of MFB with and without NAD supplement in each well. The solution from the first column was serially diluted 10-fold until the tenth column. The eleventh and twelfth column stayed sterile and were used as negative controls. The plates were incubated for 2 weeks. The number of wells in the last three columns showing color change were counted, and this number was then converted into the most probable number (M.P.N) of mycoplasmas present [9].

### 4.3. Sequencing of the MS Strains

The five isolates (18DW, 51SH, BS4S2, G3, and A4) were incubated in MFB with NAD supplement to obtain the proper amount of DNA required for sequencing. Fifty milliliters of incubated broth were used to extract DNA from each sample.

For Illumina sequencing, QiaAmp Mini Kit (Qiagen GmbH, Hilden, Germany) was used to extract the DNA. TruSeq Nano DNA Sample Preparation Kit (Illumina, San Diego, CA, USA) was used for library preparation according to the manufacturer’s instructions. Illumina sequencing was performed on the Illumina NextSeq 500 platform, resulting in 151 bp paired-end reads. The A4 strain was sequenced previously. TruSeq Nano DNA Sample Preparation Kit (Illumina, San Diego, CA, USA) was used for library preparation according to the manufacturer’s instructions for sequencing the A4 strain. Then, Illumina sequencing was performed on the HiSeq 4000 system, resulting in 101 bp, paired-end reads.

For ONT, MagAttract HMW DNA Kit (Qiagen GmbH, Hilden, Germany) was used to extract the DNA for long read sequencing according to the manufacturer’s instructions. No shearing or size selection of the DNA was performed. The ONT sequencing was performed by following the native barcoding genomic DNA protocol (with EXP-NBD104, EXP-NBD114 and SQK-LSK109). Sequencing was performed with the flow cell type R 9.4.1 on the MinION Mk1B (Oxford Nanopore Technologies, Oxford, UK) for 48 h. Albacore v 2.3.1 (Oxford Nanopore Technologies, Oxford, UK) was used for basecalling. Default parameters were used for all software unless otherwise specified.

### 4.4. Assembly of the MS Genomes

Nanopore sequencing reads were subsampled with 100 × coverage using Filtlong v 0.2.1 (University of Melbourne, Victoria, Australia) with expected genome size of 0.8 mbp. The filtered reads were assembled using the Flye v2.7.1 (University of California, San Diego, CA, USA) [10] with parameter; -size 0.8 m. The assembled draft genome was polished with the Illumina sequencing reads using unicycler_polish, which is part of the Unicycler pipeline [11]. Circularization of the assembled genome was double checked by visualization using Bandage v.0.8.1 (University of Melbourne, Victoria, Australia) [21].

### 4.5. Annotation of the Assembled Genomes

The assembled genomes were annotated using Prokka v.1.14.5 (https://github.com/tseemann/prokka accessed on 22 September 2021), a rapid prokaryotic genome annotation tool [14]. Annotation was performed with translation Table 4. Option; -gcode 4.

### 4.6. Analysis Tools Used for the Comparative Analysis of the MS Genomes and Their Parameters

BRIG v 0.95 [15], Roary v3.11.2 [16], MegaBLAST (National Center for Biotechnology Information, Bethesda, MD, USA) [22] and KEGG mapper (Institute for Chemical Research, Kyoto University, Kyoto, Japan) [19] were used to analyze the genomes. Default parameters were used for all software unless otherwise specified.

BRIG was operated using the Prokka output files in the fna format. Roary, the pan genome pipeline, was operated with the Prokka output files in the gff3 format. The Roary pipeline was performed with translation Table 4. The gene_absence_presence table was used to identify the candidates for the unique sequences present only in the genomes of NAD-independent MS isolates. MegaBLAST was performed with the Geneious v 2019.1. using NCBI nucleotide collection (nr/nt) database with the 10 candidates selected from the gene_absence_presence table. Only three unique CDSs were confirmed to be absent in the genomes of MS-H and WVU1853 strains. The KEGG mapper was applied to investigate the NAD metabolism of the MS genomes. Faa format files of the Prokka output were submitted to KEGG mapper with tool Assign KO [18] and the dataset of the genus *Mycoplasmopsis*, Taxonomy ID; 2767358 was selected.

## 5. Conclusions

Nicotinate phosphoribosyl transferase, glycerol-3-phosphatase acyltransferase and unknown CDS are found on the genomes of NAD-independent MS, *Mycoplasma synoviae*. NAPRT is thought to be the factor that brings NAD-independent trait to MS.

## Figures and Tables

**Figure 1 pathogens-10-01231-f001:**
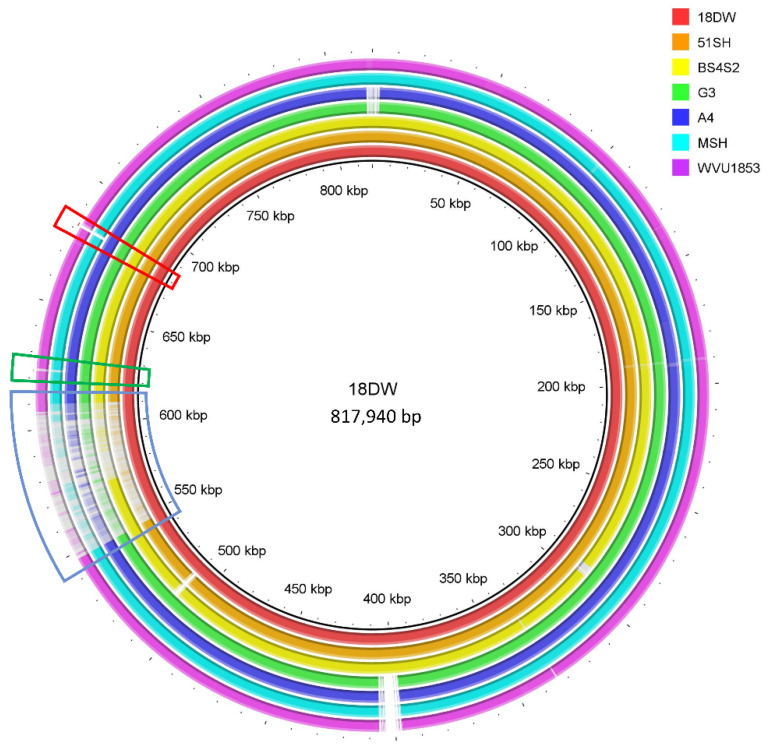
Comparison of the genomes of seven different MS strains. The differences that distinguish NAD-independent strains from NAD-dependent strains are indicated by red and green boxes. The unique coding sequences (CDSs), which will be described later, were present inside the red and green boxes. The CDSs of nicotinate phosphoribosyltransferase and glycerol-3-phosphatase acyltransferase were found to be present in the red box, while unique hypothetical protein CDS was present in the green box. The blue box indicates the vlhA pseudogene region.

**Figure 2 pathogens-10-01231-f002:**
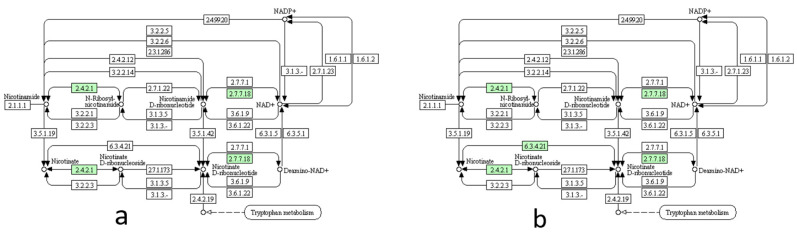
Metabolic pathways of nicotinate and nicotinamide in the MS strains, (**a**) WVU1853 and (**b**) 18DW, as reconstructed using KEGG mapper. The green boxes indicate the enzymes present in the genome of each MS strain.

**Table 1 pathogens-10-01231-t001:** Number of color-changing units (CCU) of the Mycoplasma synoviae (MS) strains in Modified Frey’s Broth (MFB) with and without nicotinamide adenine dinucleotide (NAD).

Strain	CCU in MFB without NAD ^1^	CCU in MFB with NAD ^1^	CCU Ratio ^2^
18DW	7.74 × 10^6^	2.98 × 10^6^	2.60
51SH	9.40 × 10^5^	7.74 × 10^5^	1.21
BS4S2	3.01 × 10^7^	4.45 × 10^6^	6.76
A4	1.59 × 10^7^	6.22 × 10^6^	2.56
G3	1.03 × 10^7^	1.20 × 10^7^	0.86
WVU1853	No growth	9.82 × 10^7^	0
MS-H	No growth	7.74 × 10^6^	0

^1^ CCU in 22.5 µL of the suspensions. ^2^ CCU in MFB without NAD/CCU in MFB with NAD.

**Table 2 pathogens-10-01231-t002:** Coverage of the sequencing reads and the brief assembly result of each strain.

Strain	Coverage of the Illumina Reads ^1^	Coverage of the ONT Reads ^1^	N_50_ of the ONT Reads ^2^	Number of Assembled Contig	Circularity
18DW	1962 ×	3517 ×	16,143	1	Circular
51SH	2007 ×	908 ×	5000	1	Circular
BS4S2	2083 ×	362 ×	10,545	1	Circular
G3	2178 ×	3267 ×	7352	1	Circular
A4	2974 ×	3462 ×	15,963	1	Circular

^1^ Estimated genome size was 0.8 mbp. ^2^ N_50_ is defined as the sequence length of the shortest contig at 50% of the total genome length.

**Table 3 pathogens-10-01231-t003:** General characteristics of the assembled genomes defined using Prokka.

Characteristics	Korean Isolates ^1^	WVU1853	MS-H
Total length (bp)	795,375	846,495	818,848
G + C contents (%)	28.2	28.3	28.2
Number of			
CDS	703	739	707
rRNA	7	7	7
tRNA	34	34	34
tmRNA	1	1	1

^1^ General characteristics of the assembled genomes are represented by their average or mode. The total length and G + C contents are presented by their average. The number of CDS and RNAs are represented by their mode.

**Table 4 pathogens-10-01231-t004:** BLAST results of three CDSs that were unique to the genomes of NAD-independent MS strains.

Name of CDS According to Prokka	Name of CDS on the BLAST Result	Length (bp)	Presented Genomes ^1^
CDS; Hypothetical protein	Not annotated	291	CP034544, CP079705
CDS; Hypothetical protein	Nicotinate phosphoribosyltransferase	1011	CP034544, CP079705, CP045542
plsY_2	Glycerol-3-phosphate acyltransferase	306	CP034544, CP079705

^1^ Accession numbers of the BLAST results that matched the unique CDS.

**Table 5 pathogens-10-01231-t005:** Information of MS strains used in this study.

Strain	Source of Data	Country of Origin	Accession Number
18DW	This study	Korea	CP082193
51SH	This study	Korea	CP082192
BS4S2	This study	Korea	CP082194
G3	This study	Korea	CP082195
A4	This study	Korea	CP082196
MSH	GenBank	Australia	CP011096
WVU1853	GenBank	USA	CP021129

## Data Availability

The complete genomes of 18DW, 51SH, BS4S2, G3 and A4 are available from the GenBank, NCBI under accession number, CP082192 to CP082196, BioSample; SAMN20926822 to SAMN20926826, BioProject; PRJNA756821.

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
