# Peer review of "Genome Analysis of Nicotinamide Adenine Dinucleotide-Independent Mycoplasma synoviae Isolates From Korea"

_pathogens, 2021, doi:10.3390/pathogens10101231_

Round 1

Reviewer 1 Report

The paper by Song et al, aims to decipher the genetic determinism of the NAD-dependence/independence of Mycoplasma synoviae. They isolated 5 Korean strains, evidenced their NAD-independence in vitro and sequenced their genome. Comparison of these genomes with those of two NAD-dependent strains lead to the identification of 3 CDS that are not present in the genome of the NAD-dependent strains. One of these genes is involved in the NAD metabolism.

The paper is well written and the data presented support the conclusion. Although the authors answered to the genetic determinism of the NAD-dependence of MS strains, they are not going deeply into the genomic comparison as the title suggested. For example, Figure 1 clearly shows deletion around 400 Kbp between A4/MSH/WVU153 and G3/BS4S2/51SH/18DW. Other mutations between strains are clearly showed in Figure 1. Moreover, no conclusion is drawn regarding the two other CDS absents in NAD-dependent strains.

Despite this remark, I have only several minor comments:

Keywords and lines 23-24 and line 257 : Please replace "Mycoplasmopsis" by "Mycoplasma". This name "Mycoplasmopsis" is not approved by taxonomic committee and has been rejected by several authors (https://www.microbiologyresearch.org/content/journal/ijsem/10.1099/ijsem.0.003632). I know that this name is used by several culture collections and databases. However, without official approbation, it should not be used.

Table 1: for the G3 strains: 12.0 x 106 instead of 1.20 x 107

Line 56: "sequences" instead of "seqeunces"

Line 68: please define >Q10%

Line 96: "shown" instead of "showin"

Line 111: "using" instead of "usinn"

Figure 2: the legend of figure 2 has to be completed with the meaning of the green boxes.

Results section 2.4.2: Strains HN01 and FJ-01 have to be introduced in this section or into the introduction for clarity purpose.

Materials and methods:

Section 4.2: please indicate the final concentration of NAD into the MFB.

Line 249:"operated" instead of "oprated"

Figure 1 legend: "The blue box indicates the vlhA pseudogene region". This feature is not indicated in the main text. So, is this information worth to mention it?

Author Response

Thank you for the precise review. Please see the attachment.

Reviewer 2 Report

The manuscript „Genome analysis of nicotinamide adenine dinucleotide-independent Mycoplasma synoviae isolates from South Korea” describes the comparative genome analysis of NAD-dependent and NAD-independent M. synoviae strains and reveals the presence of three coding sequences specific for the NAD-independent strains.

The Authors use current technique and widely used databases and software for genome sequencing and analysis. Advances, usefulness and limitations of the results are discussed also.

Comments:

  1. Introduction, line 23: reference citing new nomenclature is lacking, although according to Balish and co-workers (2019) and Janda (2020), this new nomenclature may not be supported at all
  2. Discussion, lines 168-180: different databases contain somewhat different data and pathways, it may worth to search de novo pathways for synthesizing NAD in other databases (e.g. Reactome, WikiPathways, MetaCyc) also
  3. Reference No. 9 doesn’t seem valid for CCU determination
  4. Figure 2 legend: explanation of colour codes is missing
  5. Table 1 – CCU in MFB without NAD column, 107 instead of 106 for strains BS4S2, A4, G3?
  6. Please resolve all abbreviation when first mentioned (line 68, N50)
  7. Spelling mistakes:
    1. line 65: „The percentatge of bases with a quality score of 30” percentage
    2. line 96: „MS strains were identified and are showin in Figure 1” shown
    3. line 100: „will be described latter were present inside the red and green boxes” later
    4. line 106: „to acquire additional clues explaing” explaining
    5. line 111: „MegaBLAST search was performed usinn the nr/nt” using
    6. line 249: „pipeline, was oprated with the Prokka ouput files” operated

Author Response

Thank you for the detailed review. Please see the attachment.
